# Seven Years Leptospirosis Follow-Up in a Critical Care Unit of a French Metropolitan Hospital; Role of Real Time PCR for a Quick and Acute Diagnosis

**DOI:** 10.3390/jcm9093011

**Published:** 2020-09-18

**Authors:** Olivier Bahuaud, Adeline Pastuszka, Cécile Le Brun, Stephan Ehrmann, Philippe Lanotte

**Affiliations:** 1Service de Bactériologie-Virologie, CHRU de Tours, 37044 Tours, France; olivier.bahuaud@gmail.com (O.B.); adeline.pastuszka@chu-tours.fr (A.P.); C.LEBRUN@chu-tours.fr (C.L.B.); 2Service de Médecine Intensive Réanimation, INSERM CIC 1415, CRICS-TriggerSep Research Network, F37044 Tours, France; stephanehrmann@gmail.com; 3UMR1282 Infectiologie et Santé Publique, INRA-Université de Tours, F-37380 Nouzilly, France; 4Centre d’étude des pathologies respiratoires, INSERM U1100, Université de Tours, F37000 Tours, France

**Keywords:** leptospirosis, intensive care unit, polymerase chain reaction, diagnosis

## Abstract

(1) Background: Leptospirosis infection can lead to multiple organ failure, requiring hospitalization in an intensive care unit for supportive care, along with initiation of an adapted antibiotic therapy. Achieving a quick diagnosis is decisive in the management of these patients. (2) Methods: We present here a review of leptospirosis cases diagnosed in the intensive care unit of our hospital over seven years. Clinical and biological data were gathered, and we compared the differences in terms of diagnostic method. (3) Results: Molecular biology method by Polymerase Chain Reaction (PCR) allowed quick and reliable diagnosis when performed in the first days after the symptoms began. Moreover, we identified that sampling blood and urine for PCR was more efficient than performing PCR on only one type of biological sample. (4) Conclusions: Our results confirm the efficiency of PCR for the quick diagnosis of leptospirosis and suggest that testing both blood and urine early in the disease might improve diagnosis.

## 1. Introduction

Leptospirosis is a widespread bacterial zoonosis caused by pathogenic Leptospira species. It is responsible for large epidemics in endemic tropical areas. The global burden of Leptospirosis is estimated at one million cases with approximately 60,000 deaths annually, occurring primarily in rural and peri-urban areas of tropical regions [1]. Although the impact of the disease is lower in developed countries, metropolitan France has one of the highest incidence rates among these countries [2]. Humans serve as accidental hosts and small mammals, mainly rodents, constitute the major reservoir. Leptospira colonize their urinary tract, thus inducing urinary shedding that can last for long periods without symptoms. Infection occurs either through direct contact with infected animals, or indirectly via mucous membranes or skin lesions that have been in contact with soil or water contaminated with urine from infected animals [3,4].

Clinical features of leptospirosis regroup a broad spectrum of symptoms [4]. Patients typically present with brutal onset of fever and non-specific symptoms such as chills, muscle pain and headache [3,4]. The typical association of jaundice and kidney failure is known as Weil’s disease. Conjunctival suffusion is also common during leptospirosis [5]. Pulmonary and/or abdominal symptoms are often involved in these patients, including severe cough, dyspnea, cholecystitis and pancreatitis. Thrombocytopenia occurs frequently and although it does not usually cause spontaneous bleeding, patients can develop severe gastrointestinal or pulmonary hemorrhage [4]. Leptospirosis can also be responsible for severe infections leading to multiple organ dysfunctions including brain, lung, liver and kidney failure. In these cases, admission to an Intensive Care Unit (ICU) and provision of vital support is required, along with an adapted antimicrobial therapy such as aminopenicillin, third generation cephalosporins or doxycycline [6].

Diagnosis is confirmed after biological identification of the pathogen. Conventional diagnosis was initially based on culturing and observing Leptospira by dark field microscopy; however, given the fastidious culture characteristic, direct identification has been replaced by serological diagnosis based on detection of anti-Leptospira antibodies by Microscopic Agglutination Test (MAT), considered a gold standard technique [7]. However, the delay between the onset of the disease and the detectability of the anti-Leptospira antibodies represents a downside of this method. Indeed, the serological confirmation often comes well after the beginning of the symptoms, which is in contrast with the necessity of an accurate diagnosis, which would allow the initiation of an adapted therapy as early as possible. Molecular detection methods now allow detection and genotyping of the pathogen in the early phase of the disease [8,9,10,11,12]. Current guidelines recommend testing either blood, urine or cerebrospinal fluid (CSF) depending on the delay since the first symptoms [6,7]. Several quantitative PCR (Polymerase Chain Reaction) assays have been described that target different genes [9,13,14,15,16,17]. Merien et al. have designed a PCR assay that targets the specific gene *lfb1* (encoding leptospira fibronectin binding protein 1), conserved among the pathogenic leptospira species, allowing early diagnosis from blood samples [17]. Our laboratory uses this *lfb1* PCR assay to diagnose leptospirosis in patients. Here we present a review of leptospirosis cases, diagnosed by a molecular approach using the *lfb1* PCR on blood, urine and/or CSF samples between January 2011 and January 2018 in the University Hospital of Tours, France.

## 2. Experimental Section

### 2.1. Study Design

We carried out a monocentric review of severe leptospirosis cases hospitalized in the ICU at the University Hospital of Tours (France) occurring between January 2011 and January 2018. Patients presenting with clinical, biological and/or epidemiological data evocative of leptospirosis, associated with a positive molecular or serological testing, were considered as confirmed cases and included in this study.

### 2.2. Real-Time PCR Method

The real-time PCR method was adapted from Merien et al. and Bourhy et al. [9,17]. The in-house PCR assay targeted the *lfb1* gene. Samples were either blood collected on EDTA, urine or CSF, or tissue from a solid organ biopsy performed during autopsy. Pretreatment of the samples was required for DNA extraction from urine, blood, tissues and CSF samples. For urine samples, 1 mL of urine was centrifugated for 5 min at 3000 rpm, the supernatant was discarded, and the pellet was resuspended by adding 1 mL PBS (Sigma Aldrich, Saint Louis, USA). After a new step of centrifugation, the supernatant was discarded and the pellet was resuspended by adding 200 µL G2 (Qiagen, Hilden, Germany). G2 buffer is a general lysis buffer containing guanidine hydrochloride, tween and triton X100. For CSF, tissues and plasma obtained from blood samples, 200 µL of samples were added to 190 µL G2 buffer and 10 µL lysozyme (100 mg/mL). A step of 10 min at 37 °C was necessary before DNA extraction. DNA was extracted from 200 µL of pretreated sample using EZ1 instruments (Qiagen, Germany) according to the manufacturer’s instructions. DNA was then eluted under a volume of 100 µL. We used non-frozen DNA obtained from fresh samples for the analysis.

Real time PCR was performed in a Smart cycler^®^ (Cepheid, Sunnyvale, USA) with 5 μL of DNA, 12.5 µL of Master Mix Premix Ex Taq (Takara, Saint-Germain-en-Laye, France), 2.5 µL of Sybr Green (Sigma, France), 1 µL of each primer at 10 µM (LFB1-F 5′-CATTCATGTTTCGAATCATTTCAAA-3′ and LFB1-R 5′-GGCCCAAGTTCCTTCTAAAAG-3′) and 3 μL of water. Thermal cycling conditions were as follows: 1 cycle of 30 s at 95 °C followed by 45 cycles of 15 s at 95 °C, 30 s at 58 °C and 30 s at 72 °C. The last step was a ramp up from 58 °C to 95 °C by 0.2 °C/s to obtain a melting peak curve. Beta-globulin detection was performed on all samples to test for the presence of inhibitors. Our method is regularly subject to performance review via external quality control from the French National Reference Center for Leptospira (Pasteur Institute, Paris, France).

### 2.3. Clinical and Biological Data

Over the study period the cases were identified using the database of the Program for Medicalization of Information System (PMSI) looking for confirmed cases of leptospirosis in patients hospitalized in the Intensive Care Units of our Hospital. Data were collected retrospectively through the computerized medical files of the patients using the hospital medical software: DXlab^®^ (v4.23.30, MEDASYS, Le Plessis-Robinson, France) and Millenium^®^ (v2015.01.19, CERNER, Paris-La défense, France). The data analyzed were age and sex of the patients, clinical and biological symptoms suggestive of leptospirosis and date of events (year, month). Results from molecular biology and serological testing, performed by EIA completed with MAT, were collected and reported for each patient in addition to the outcome of the disease and antimicrobial therapy used.

## 3. Results

### 3.1. Epidemiological Data

Over a seven-year period (January 2011 to January 2018) the PMSI database of Tours University Hospital identified 16 cases of confirmed leptospirosis in adult patients hospitalized in the ICU. Our laboratory received 75 samples of serum, urine and CSF from 34 patients hospitalized in the ICU to test for leptospirosis with a molecular technique. We performed our in-house PCR assay targeting *lfb1* on these samples. Among the 34 patients, 15 (44%) were found positive for leptospirosis involving 33 (44%) of the 75 samples tested. Only one patient showed negative PCR results but positive serological results. 

The median age at diagnosis was 56 years (Table 1). Sex ratio was noticeably in favor of male with 14 out of 16 patients (87%). The majority of cases (13/16; 81%) occurred over the summer period from May to September including a peak of 6 cases in July. Only 3 cases (19%) occurred over the winter period from November to January. The source of contamination was detected in almost every case. The main source of exposure was bathing and swimming in rivers (5 patients, 31%). The other included having contact with or drinking stagnant water (4 patients, 25%), fishing (3 patients, 19%), hunting (3 patients, 19%) (one patient having been exposed through both fishing and hunting) and gardening (1 patient, 6%). For one patient the source of contamination was not identified, even postmortem.

### 3.2. Clinical and Biological Characteristics

Although the clinical presentation was often typical, especially among severe cases when ICU admission was needed, we noticed disparities in the clinical features (Table 1). All patients (16/16) presented with fever and thrombocytopenia. The majority (15/16; 94%) presented with kidney failure, including one patient who required renal replacement therapy. Jaundice was observed in 13 patients (81%) and 8 (50%) showed elevated liver enzymes. The characteristic association of jaundice and kidney failure defined as Weil’s disease was observed in 12 cases (75%). There were also 12 patients (75%) describing arthralgia and myalgia but only 4 presented (25%) with rhabdomyolysis. Acute respiratory failure was observed in 6 patients (37%) and digestive disorders including diarrhea and vomiting in 5 patients (31%).

The outcome was favorable in 14 out of the 16 cases (87%) with complete recovery following adequate antimicrobial therapy. Among these 14 patients, 8 patients (57%) had been treated with ceftriaxone only, while 3 patients (19%) were switched to second-line amoxicillin per os and 2 (12%) received second-line oral therapy with doxycycline. One patient received amoxicillin followed by doxycycline. Each of these 14 patients completed a course of at least 10 days of antibiotic therapy. The other 2 patients (12%) died of multiple organ failure and although they received ceftriaxone, death occurred before 10 days of treatment.

### 3.3. Leptospirosis Sampling Results

Samples from the 16 patients have been tested with our in-house PCR assay targeting *lfb1*. Among them, 14 patients had both urine and blood samples collected while 2 patients only had one collected. We divided the patients into three distinct groups according to the in-house PCR assay results on blood and urine samples. Eight patients (50%) tested positive on both blood and urine. The median duration between first symptoms and positive results in this group was 4.5 days (±2.5). Four patients (25%) tested positive only on the urine samples while 3 (19%) tested positive only on the blood samples. Median duration between first symptoms and positive results in these groups was 6 days (±2) and 4.5 days (±0.5), respectively.

Considering the samples independently, PCR was positive on blood in 11 patients and on urine in 12 patients (out of the 15 patients showing at least one positive PCR result (Figure 1)). The median duration between the beginning of symptoms and positive result of PCR on blood samples was 5 days (range 2–7) versus 6 days (range 2–8) on urine samples.

Among the 16 patients, 11 (69%) had serological testing for leptospirosis. Median duration between first symptoms and sampling was 7 days (range 2–30). Four patients presented with positive serology on the first sample and one presented with seroconversion over a period of 15 days, 6 patients showed negative results. Median duration between first symptoms and sampling was 17 days (range 6–30) for positive results and 6 days (range 2–8) for negative results. One patient was diagnosed with serology only. Although blood, urine and CSF samples from this patient had been tested by PCR 10 days after the beginning of the symptoms, PCR results were negative on all the samples. The CSF in this case was probably tested by excess because patient presented headache without a true meningitis syndrome. However, serology performed on blood on Day 17, after the first symptoms, allowed the diagnosis of leptospirosis.

## 4. Discussion

Leptospirosis remains a worldwide public health concern with many cases each year. Even though its burden is higher in tropical and subtropical areas, developed countries have experienced major outbreaks or abnormally high prevalence over the last decade. Many fear that rapid urbanization in developing countries and global warning, along with natural disasters (flooding, massive earthquake, hurricanes, etc.), will contribute to the spread of this disease. Moreover, numerous reports mention the extension of the animal reservoir in different locations worldwide [18,19] as well as the identification of new strains of Leptospira including potentially pathogenic strains [20]. This emphasizes the need for monitoring and prevention to achieve disease control.

Efforts should also be focused on diagnosis. Indeed, life-threatening forms of leptospirosis rapidly progress from fever to multiple organ failure. The cases described in this study presented severe forms of leptospirosis. This can be explained by the recruitment of patients hospitalized in the ICU. However, 87% experienced full recovery after receiving adequate antibiotic therapy along with supportive care initiated rapidly after diagnosis. This underlines the importance of a quick and accurate diagnosis to give the patient the best chances of recovery.

Several works have presented the successful use of PCR in early diagnosis of leptospirosis [9,10,11,18]. The findings of this study are consistent with those studies. Indeed, only five (31%) patients presented positive serological results among those tested with a median duration between first symptoms and positive results of 17 days. Whereas, as expected, infected patients with negative serology results presented a median duration of 6 days between first symptoms and sampling. 

However, the majority of our patients (15 out of 16, 94%) were diagnosed early through PCR. Indeed, median duration between onset of symptoms and positive results of PCR was 4.5 days. When taken independently, these durations were 5 and 6 days for blood and urine samples, respectively. The samples of the only patient with negative PCR results were collected 10 days after the onset of the symptoms, which may explain the negativity. This delay of positivity is concordant with results found in previous studies [10,21]. Thus, even though the number of patients is limited, it appears that PCR shows a higher diagnostic sensibility than serological testing with 94% against 45% of positive results. These results need to be confirmed on a larger number of patients, but they emphasize the potential of molecular testing through PCR for an early diagnosis, in comparison to serological testing. 

Interestingly, only eight patients were found to have positive results for both urine and blood samples while four were diagnosed through urine and three through blood sample positivity only. Performing PCR on samples from both anatomical sites (blood and urine) allowed the diagnosis of 15 patients, while only 11 patients would have been diagnosed if only tested on blood and 12 if only tested on urine (Figure 1). These results are in agreement with a previous work from Esteves et al. where main cases were diagnosed at an early stage of the infection corresponding to dissemination and kidney colonization [22]. Thus, sampling on only one site would have left respectively 4 or 3 patients, respectively, with negative PCR results. Even though the number of patients was low, these results suggest that sampling both sites for each patient improved the number of leptospirosis cases diagnosed.

Persistent shedding of Leptospira in the urine of the infected hosts is well known [3,4]. Diagnosis of leptospirosis through PCR on urine samples has already been described and is often used in animals for monitoring purposes [16,23,24]. However, some recent studies in humans showed only PCR results on blood samples [10,21,25] while acknowledging the potential of urine sampling to improve sensibility [21]. A recent study on leptospirosis cases in 79 French Metropolitan ICUs showed that 53% of patients were diagnosed by PCR on blood while only 11% were diagnosed by PCR on urine samples [26]. Even though the number of PCR tests performed on the different type of samples was not indicated, this study suggests that urine sampling was not as frequent as blood sampling for PCR testing. In this context, our study, consistently with others, underlines the interest of collecting both urine and blood samples early in patients with symptoms of leptospirosis. 

## 5. Conclusions

In conclusion, we highlight in this study the importance of early diagnosis of leptospirosis, which can be obtained with great sensitivity through PCR. We also underline that collecting urine samples along with blood samples might improve the sensitivity and provide a more accurate diagnosis. Further studies are needed to precisely determine the optimal time of sampling blood and urine to confirm diagnosis of leptospirosis. Earlier seems to be best for blood samples, and urine may shift in time. Nevertheless, both seem unavoidable.

## Figures and Tables

**Figure 1 jcm-09-03011-f001:**
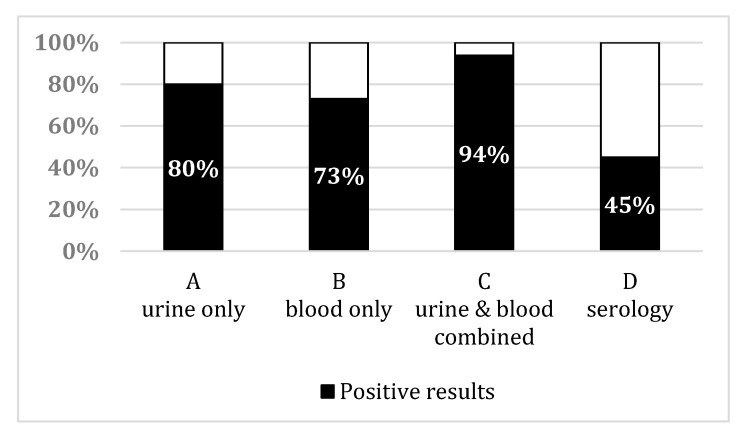
Percentage of leptospirosis diagnosis confirmed in our patients based on positive PCR results among: (**A**) Urine samples only, (**B**) blood samples only, (**C**) when combining urine and/or blood for each patient. The percentage of positive serology results among all serological samples is given in (**D**) as a comparison.

**Table 1 jcm-09-03011-t001:** Main characteristics of patients hospitalized in ICU with a confirmed leptospirosis diagnosis.

Age	Gender	Date (M/Y)	Main Symptoms	Biological Data	Samples (Results)	Delay between First Symptom and Sampling (Days)	Duration of ICU Hospitalization (Days)	Treatment Received	Outcome	Epidemiological Data
60	M	September 2011	Fever, myalgia; acute respiratory distress and acute kidney failure	Creatininemia: 516 µmol/L	PCR on blood and urine (positive)/no serology performed	3	1	Ceftriaxone	Death	Contact with stagnant water
				Elevated liver enzymes						
				Thrombocytopenia: 81 G/L						
60	F	April 2012	Fever; vomiting; acute respiratory distress syndrome due to intra-alveolar hemorrhage, acute kidney failure and jaundice	Creatininemia: 405µmol/L	PCR on blood, urine, lung and liver post mortem biopsies (positive)/serology on day 9 (positive)	7	2	Ceftriaxone	Death	Unknown
				Elevated liver enzymes						
				Bilirubinemia: 341µmol/L						
				Thrombocytopenia: 8 G/L						
55	M	July 2013	Fever; myalgia, acute respiratory distress; acute kidney failure and jaundice	Creatininemia: 168 µmol/L	PCR on blood and urine (positive)/serology on day 6 (negative)	6	3	Ceftriaxone	Complete recovery	Swimming in a river
				Bilirubinemia: 706 µmol/L Thrombocytopenia: 12 G/L						
20	M	August 2013	Fever; myalgia, acute respiratory distress with intra alveolar hemorrhage; acute kidney failure and jaundice	Creatininemia: 124 µmol/L	PCR on blood and urine (positive)/serology on day 2 (negative)	2	8	Ceftriaxone	Complete recovery	Swimming in a river/skin lesions
				Bilirubinemia: 217 µmol/L Thrombocytopenia: 25 G/L						
43	M	June 2014	Fever, myalgia, arthralgia, diarrhea, cutaneous rash; Acute Kidney Failure	Creatininemia: 140 µmol/L	PCR on blood (positive) and urine (negative)/serology on day 7 (negative)	4	8	Ceftriaxone followed by Amoxicillin	Complete recovery	Consumption of stagnant water
				Elevated liver enzymes						
				Rhabdomyolysis						
39	M	July 2014	Fever and jaundice	Bilirubinemia: 53 µmol/L	PCR on blood (positive) no urine sampling/serology on day 8 (negative)	7	8	Ceftriaxone followed by Doxycycline	Complete recovery	Swimming in a river
				Elevated liver enzymes Thrombocytopenia: 87 G/L						
49	M	August 2014	Fever, myalgia, diarrhea, acute respiratory distress with intra-alveolar hemorrhage requiring oro-tracheal intubation; acute kidney failure and jaundice	Creatininemia: 223 µmol/L	PCR on urine (positive) no plasma sampling/serology on day 7 (negative) and day 22 (positive)	7	14	Ceftriaxone	Complete recovery	Fishing
				Elevated liver enzyme						
				Bilirubinemia: 405 µmol/L						
				Anemia						
				Thrombocytopenia: 10 G/L						
71	F	August 2014	Fever, diarrhea, acute kidney failure and jaundice	Creatininemia: 164 µmol/L	PCR on blood and urine (positive)/no serology performed	4	6	Ceftriaxone followed by Doxycycline	Complete recovery	Contact with stagnant water and consumption of water from a well
				Elevated liver enzymes						
				Bilirubinemia: 278 µmol/L						
				Thrombocytopenia: 57 G/L						
65	M	November 2014	Fever, acute kidney failure and jaundice	Creatininemia: 411 µmol/L	PCR on blood and urine (positive)/serology on day 30 (positive)	6	3	Ceftriaxone followed by Amoxicillin	Complete recovery	Hunting
				Elevated liver enzymes						
				Bilirubinemia: 46 µmol/L						
				Thrombocytopenia: 10 G/L						
58	M	December 2014	Fever, myalgia, arthralgia, acute kidney failure and jaundice	Creatininemia: 327 µmol/L	PCR on blood, urine and CSF (negative)/serology on day 17 (positive)	10	4	Amoxicillin and Ofloxacin	Complete recovery	Swimming in a river
				Elevated liver enzymes						
				Bilirubinemia: 139 µmol/L						
				Thrombocytopenia: 83 G/L						
37	M	May 2015	Fever, myalgia, rhabdomyolysis, acute kidney failure	Creatininemia: 210 µmol/L	PCR on blood (negative) and urine (positive)/serology on day 6 (positive)	6	4	Ceftriaxone	Complete recovery	Swimming in a river
				Elevated liver enzymes						
				Thrombocytopenia: 139 G/L						
72	M	June 2015	Fever, myalgia, acute kidney failure requiring extra-renal purification, and jaundice	Creatininemia: 404 µmol/L Elevated liver enzymes Bilirubinemia: 177µmol/L Thrombocytopenia: 12 G/L Rhabdomyolysis	PCR on blood and urine (positive)/serology on day 3 (negative)	3	8	Ceftriaxone	Complete recovery	Hunting
61	M	July 2015	Fever, myalgia, acute respiratory distress requiring oro-tracheal intubation, acute kidney failure with hematuria, and jaundice	Creatininemia: 656 µmol/L	PCR on blood and urine (positive)/no serology performed	5	9	Ceftriaxone followed by Amoxicillin	Complete recovery	Hunting and fishing
				Elevated liver enzymes						
				Bilirubinemia: 406 µmol/L						
				Thrombocytopenia: 15 G/L						
				Anemia						
51	M	July 2015	Fever, myalgia, arthralgia, acute kidney failure and jaundice	Creatininemia: 257 µmol/L	PCR on blood (negative) and urine (positive)/serology on day 6 (negative)	6	3	Ceftriaxone	Complete recovery	Fishing
				Elevated liver enzymes						
				Bilirubinemia: 31 µmol/L Thrombocytopenia: 102 G/L						
15	M	September 2016	Fever, myalgia, arthralgia, acute kidney failure and jaundice	Creatininemia: 137 µmol/L	PCR on blood (positive) and urine (negative)/no serology performed	5	4	Ceftriaxone	Complete recovery	Swimming in a river
				Elevated liver enzymes						
				Bilirubinemia: 52 µmol/L Thrombocytopenia: 22 G/L						
62	M	January 2017	Fever, myalgia, arthralgia, rhabdomyolysis, acute kidney failure and jaundice	Creatininemia: 331 µmol/L	PCR on blood (negative) and urine (positive)/no serology performed	8	2	Ceftriaxone	Complete recovery	Gardening

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
