# Peer review of "Seven Years Leptospirosis Follow-Up in a Critical Care Unit of a French Metropolitan Hospital; Role of Real Time PCR for a Quick and Acute Diagnosis"

_jcm, 2020, doi:10.3390/jcm9093011_

Round 1

Reviewer 1 Report

The paper describes the application of PCR for diagnosis of leptospirosis.

Items that are not fully described are listed below.

The criteria of diagnosis for patients should be given.

Detailed methods for serological diagnosis should be described.

Line 31. Conclusions; Why did they draw conclusion multiple sampling is beneficial. Time of sampling or site of sampling.  Which data supports this conclusion.

Line 80-81; samples from blood, tissues, urine.  Did they use fresh samples or stored samples for their analysis?

Blood is whole blood? There are no data for tissues or CSF. Why they did not centrifuge CSF like urine?

Line 85 what is G2.

Reference of the methods for extraction of DNA should be given.

Finally, the paper describes on the results of PCR and more description of clinical aspects would be necessary.

Author Response

Answer to Reviewer n°1:

  • “The criteria of diagnosis for patients should be given. “
    • To answer to this point a definition has been added in the 2.1 section (lines 81-84 of the revised manuscript).

  • Detailed methods for serological diagnosis should be described.
    • to answer to this point a sentence has been added in the 2.3 section : “… and serological testing, performed by EIA completed with MAT, were collected….) (lines 143-144 of the revised manuscript).

  • Line 31. Conclusions; why did they draw conclusion multiple sampling is beneficial. Time of sampling or site of sampling.  Which data supports this conclusion?
    • To answer to the reviewer we have modified the last sentence of the introduction section as follow :” Our results confirm the efficiency of PCR for the quick diagnosis of leptospirosis and suggest that testing both blood and urine early in the disease might improve diagnosis.” (lines 30-31 of the revised manuscript).
    • Data supporting this conclusion are presented on figure 1 and in the 3.3 section entitled leptospirosis sampling results in the text. We show in this figure that combination of PCR performed on blood and urine allow diagnosis of 94% of the cases instead of 80% on urine and 73% when PCR is performed on blood only.

We have also improved the discussion of results supporting this conclusion (lines 252-259 of the revised manuscript) and we have added a sentence at the end of the MS (lines 275-277) :”Further studies are needed to precise the optimal time of sampling blood and urine to confirm diagnosis of leptospirosis. The earlier seems to be the best for blood samples and urine may be shifted in time. Nevertheless, both seem unavoidable”

  • Line 80-81; samples from blood, tissues, urine.  Did they use fresh samples or stored samples for their analysis?
    • Fresh samples have been used. This information have been added on line 99 : “We used non frozen DNA obtained from fresh samples for the analysis. »

  • Blood is whole blood? There are no data for tissues or CSF. Why they did not centrifuge CSF like urine?
    • Plasma used for analysis was obtained from blood samples collected on EDTA. This was added on lines 88 and 95 of the revised manuscript.
    • Tissues obtained in post mortem (lung and liver) for patient 2012/04 aged of 60 years, were positive by PCR. This information was added in table 1 of the revised MS.
    • CSF was only tested for one patient who presented headache without true meningitis syndrome (CSF : 8 leukocytes/mm3). A sentence has been added on lines 216-217.
    • CSF was not centrifuged because this is not recommended usually for other pathogens.

  • Line 85 what is G2.
    • G2 buffer is a general lysis buffer containing guanidine hydrochloride, tween and triton X100. This was noted on the revised MS on lines 93-95
    •  
  • Reference of the methods for extraction of DNA should be given.
    • The method for extraction follow the manufacturer instructions. This was already noted on line 98 of the revised manuscript.
    •  
  • Finally, the paper describes on the results of PCR and more description of clinical aspects would be necessary.
    • We thought that clinical aspects have been already detailed especially in table 1 and in the text in the section 3.2 (lines 171-186 of the revised manuscript). We do not know how to answer more precisely to the reviewer on this point.

Reviewer 2 Report

The manuscript (jcm-920083) submitted by Bahuaud et al describes a nicely conducted study, very well writing with few grammatical errors, that merits publication after some small modifications, This study describes the importance of real time PCR in the early diagnosis of leptospirosis in patients hospitalized in the ICU over seven years.  For this analysis, the authors were carful to chose patients samples in the beginning of symptoms, which is crucial to an early diagnosis of leptospirosis.

Although, the number of studied patients was low, the obtained results are very interesting in a clinical perspective, since it shows blood and urine samples should both be analyzed in an early phase of leptospirosis. Therefore, and considering the kinetics of Leptospira infection and disease progression in humans, can the authors explain how it is possible to have positive results in both blood and urine samples? A brief explanation to this fact is given in the last part of the discussion but it does not explain why. Please consult this article:

“Esteves LM, Bulhões SM, Branco CC, et al. Diagnosis of Human Leptospirosis in a Clinical Setting: Real-Time PCR High Resolution Melting Analysis for Detection of Leptospira at the Onset of Disease. Sci Rep. 2018;8(1):9213. Published 2018 Jun 15. doi:10.1038/s41598-018-27555-2”

Also, the authors referred that CSF samples were analyzed but did not shows the results for this type of sample. And, in what phase of leptospirosis is the CSF important to analyze?  

Finally, I did not find point 4, which I think is where the discussion begins, probably after line 175 in page 5?

Other comments:

Page 2, line 58: remove “with”

Page 2, line 68: gene lfb1 referees to what gene?

 Page 2, line 78: I advise the authors to change “PCR method” by “real time PCR method”

Page 2, line 85: Can the authors explain what is G2 buffer?

Page 5, line 167: What was the serological test performed? Was it MAT?

Author Response

Answer to Reviewer n°2:

  • Therefore, and considering the kinetics of Leptospira infection and disease progression in humans, can the authors explain how it is possible to have positive results in both blood and urine samples? A brief explanation to this fact is given in the last part of the discussion but it does not explain why.
    • To answer to this point, we have modified the MS on lines 252-259 and have added the reference suggested by the reviewer 2 (Esteves et al, Sci Rep 2018). In this paper diagnosis were mainly performed at early stage of the infection were blood and urine were positive both.

  • Also, the authors referred that CSF samples were analysed but did not shows the results for this type of sample. And, in what phase of leptospirosis is the CSF important to analyse?  
    • Result of CSF is already present on table 1. As answered to reviewer 1 remark, CSF was only tested for one patient who presented headache without true meningitis syndrome A sentence has been added on lines 216-217 of the revised manuscript.
    • CSF should be probably interesting to test when patient present a meningitis syndrome.

  • Finally, I did not find point 4, which I think is where the discussion begins, probably after line 175 in page 5?
    • Point 4. Discussion has been added on line 219. We apologize for this missing title.

  • Page 2, line 58: remove “with”
    • We thank the reviewer for this correction. The modification has been made.

  • Page 2, line 68: gene lfb1 referees to what gene?
    • We added the information on this gene coding for the leptospira fibronectin binding protein1. Its localisation on the genome of Leptospira Interrogans and the primers used for this PCR are presented in the original article published by Merien et al (reference n°17 in our article) (lines 72-73 of the revised manuscript).

  • Page 2, line 78: I advise the authors to change “PCR method” by “real time PCR method”
    • We thank the reviewer for this advice. We performed the correction as proposed (lines 86-87)

  • Page 2, line 85: Can the authors explain what is G2 buffer?
    • This was noted on the revised MS on lines 94-95 as asked by reviewer 1

  • Page 5, line 167: What was the serological test performed? Was it MAT?
    • Serological tests were performed by EIA method confirmed by MAT. This information was added on line 143-144.

Round 2

Reviewer 1 Report

The manuscript has improved.

Following description should be give explanation

  1. Which ELISA kit was udes.
  2. Which bacteria strains were used for MAT 

Line 121 performed by EIA completed with MAT

Author Response

Answer to Reviewer n°1:

Following description should be give explanation

  1. Which ELISA kit was used.

We thank the reviewer to ask us to precise serological methods used in the study in order to improve the manuscript.

Serology used was based on EIA method (Serion ELISA classic LEPTOSPIRA IgM (Serion diagnostics)). In case of positivity, the sera have been sent to the French National Reference Center for leptospirosis where a home EIA test and a MAT were performed to confirm the positivity. 

This point was added on line 144-146 of the revised manuscript.

  1. Which bacteria strains were used for MAT 

MAT performed at the French National Reference Center for leptospirosis used Antigen from 24 strains. To answer to this question we have modified the revised manuscript as follow (lines 147-151):

Antigens used for MAT were originated from the following Bacteria: L. Australis, L. Autumnalis, L. Ballum, L. Bataviae, L. Canicola, L. Celledoni, L Cynopteri, L. Djasiman, L. Grippotyphosa, L. Hardjo, L. Hebdomadis, L. Icterohaemorrhagiae, L. Icterohaemorrhagiae strain Verdun, L. Javanica, L. Louisiana, L. Mini, L. Panama, L. patoc, L. Pomona, L. Pyrogenes, L. Sarmin, L. Sejroe, L. Shermani, L. Tarassovi.